

# Validity and reliability of the WIMU inertial device for the assessment of the vertical jump

José Pino-Ortega[1], Javier García-Rubio[2,3] and Sergio J. Ibáñez[3]

[1] Faculty of Sports Sciences, Universidad de Murcia, Spain
[2] Facultad de Educación, Universidad Autonoma de Chile, Chile
[3] Faculty of Sports Sciences, University of Extremadura, Spain

## ABSTRACT

The aim of this study was to test the validity and reliability of the inertial device WIMU (Realtrack Systems SL, Almería, Spain) for the assessment of the vertical jump, counter movement jump (CMJ) and squat jump (SJ). Fifteen soccer players were evaluated in two identical sessions separated by one week. In each session, participants performed three jumps of each type. The flight time was quantified by the inertial device WIMU and by a force platform (Twin Plates; Globus Sport and Health Technologies LLC, Codogné, Italy) at the same time. For the analysis of reliability of the flight time of the CMJ and the SJ, the intraclass correlation coefficient was used. The calculation of the concurrent validity was performed by using the Pearson correlation coefficient ($r$). This analysis was complemented with the realization of the Bland–Altman plots. For the analysis of reliability, the coefficient of variation and the standard error of the means were calculated. The analysis presented a high validity and reliability of the device. The results show the inertial device WIMU (Realtrack Systems SL, Almería, Spain) as a useful tool for measuring the jump capacity of the athletes, presenting immediate results in real time, on any type of surface and in a simple way since it does not need cables.

## INTRODUCTION

One of the most commonly used indicators to assess physical fitness in different in both the general population and high performance sports, and for identifying young talents (*Balsalobre-Fernández, Glaister & Lockey, 2015*) is the vertical jump. Similarly, it has been used to predict the risk of injury and is associated with muscle power or neuromuscular fatigue (*Jimenez-Reyes et al., 2016*; *Stevenson et al., 2015*). In fact, jump performance plays a crucial role in both the predominantly anaerobic and aerobic disciplines, as well as in the mixed disciplines (*Brumitt et al., 2014*; *Hartman et al., 2007*). This can be measured by different tools or systems (force plates, video analysis, photoelectric cells, contact mat, video cameras, etc.) (*Bui et al., 2015*; *Casartelli, Müller & Maffiuletti, 2010*; *Dias et al., 2011*) and more recently with phone apps as My Jump (*Balsalobre-Fernández, Glaister & Lockey, 2015*). Although they are highly accurate measurement systems, they involve some drawbacks. For example, force plates are difficult

Corresponding author
Javier García-Rubio, jagaru@unex.es

to use in field test while the video analysis does not generate immediate results (*Balsalobre-Fernández, Glaister & Lockey, 2015*). Similarly, video analysis has a major drawback, especially if only one camera is used. If the plane that records the camera moves, the accuracy will be affected (*Magnúsdóttir, Orgilsson & Karlsson, 2014*). There are several means for evaluating vertical jump, therefore it is important to choose the one that best fits to complete the task, regarding precision, cost, reliability or duration (*Bui et al., 2015*). Accelerometers, however, are a highly portable, lightweight, easy to use and accessible tool for almost any coach (*Casartelli, Müller & Maffiuletti, 2010*; *Sato, Smith & Sands, 2009*).

The vertical jump height is measured as the difference between the position of the center of mass of the individual in the starting position (a standing position) and its position at the maximum height (*Choukou, Laffaye & Taiar, 2014*). This height can be measured according to the flight time (*Bosco, Luhtanen & Komi, 1983*). Flight time is the raw data that both devices recorded and is used in several studies (*Balsalobre-Fernández et al., 2014*; *García-López et al., 2013*; *Garcia-Lopez et al., 2005*) and reported as the more accurate method to calculate jump height (*Aragón, 2000*). The estimation of jumping performance is done after the event. Also, the jump height is an indirect technique to measure jump height or muscle power and as (*Dias et al., 2011*) suggest, the different methods used in the calculation could be led to a systematic error. Not all the measurement systems are equally accurate; in fact, it has been reported that, due to the different angles of the knees and ankles at the time of the jump starts, there may be an error of 2.2 cm (*Garcia-Lopez et al., 2005*). WIMU$^{TM}$ is an inertial device designed for monitoring the physical activity for athletes of different disciplines.

The control of the validity and reliability of the WIMU$^{TM}$ device is necessary for its acceptance as an accurate measurement system for assessing the vertical jump. The validity and reliability refer to the device's ability to measure what it is designed to measure (validity), and to always measure the same event in the same way (reliability) (*Weir, 2005*). Therefore, the objective of this study was to evaluate the validity and reliability of the WIMU$^{TM}$ device for assessing squat jump (SJ) and counter movement jump (CMJ). It has been hypothesized that the WIMU$^{TM}$ device will display a high validity and reliability for measuring the different vertical jumps used in the study.

## MATERIALS AND METHODS

### Experimental approach to the problem

The aim of this study was to test the validity and reliability of the device WIMU$^{TM}$ (Realtrack Systems SL, Almería, Spain) to measure the performance in the vertical jump. In order to do that, the tests for measuring the flight time during a slow cycle of stretching and shortening the muscle (CMJ) and the explosive concentric muscle actions (SJ) were conducted by 15 subjects into two identical sessions. Data collections were separated by one week. Jumps were measured with the WIMU$^{TM}$ device and the force plate at the same time (Twin Plates; Globus Sport and Health Technologies LLC, Codogné, Italy), considered as the *gold standard*. The election of flight time as variable of comparison is because is the direct data that provide both systems. Jump height is an indirect

measurement of a formula based on flight time. WIMU™ was attached to a belt and fixed on the lower back (*Choukou, Laffaye & Taiar, 2014*).

## Subjects

The participants in the study were 15 soccer players at early stages ($N = 15$, age $= 14.74 \pm 0.23$ years old, height $= 164.34 \pm 4.04$ cm, weight $= 65.7 \pm 3.35$ kg). All the participants belonged to the same soccer team participating at a regional category in Spain. The team trains three sessions of 2 h per week, with a total volume of 6 h of training. The jumps were performed during the competitive season. All subjects and their parents were informed about the study procedures and their possible risks, giving their consent to participate before testing. The ethics committee of the University approved the study (nº 67/2017).

## Procedure

In both sessions of data acquisition, conditions were the same. During the week of training, the jumps were performed three days after the last game. Before jumping, warming-up was equal in both sessions. A standard warm-up of 15 min, consisting of 5 min of continuous running, specific mobility of the lower body for another 5 min, active stretching of the lower extremities, and vertical jumps for the last 5 min (*Balsalobre-Fernández, Glaister & Lockey, 2015*; *Casartelli, Müller & Maffiuletti, 2010*). Previously, the subjects had been instructed in performing the different jumps by the same examiner.

*Testing procedures.* The subjects performed each CMJ starting from a static standing position, with their arms on their hips and with their knees extended during the flight. Once in position, the subjects were instructed (approximately a 90° angle) as quickly as possible and then jump as high as possible. In SJ, subjects began in the same standing position as in the CMJ, also with their hands on their hips. In this position, they were asked to flex their hips (aprox. 90°) and maintain this position. After that, the examiner made them wait for three seconds and then cheered them up verbally to jump as high as possible, without making any type of counter movement. In all jumps at takeoff, participants were asked to leave the floor at the same time and always with their knees extended, and land in similarly extended position (*Casartelli, Müller & Maffiuletti, 2010*; *Häkkinen & Komi, 1985*). A total of six jumps were evaluated for each participant, measured with the inertial device WIMU™ and the contact plate (Fig. 1).

WIMU™ (Fig. 2). For this study, trials have been recorded by the accelerometer at a sampling frequency of 1,000 Hz. Accelerometers and gyroscopes integrated in the device were used to correct and calculate the vertical acceleration recordings. All the information is represented in its specific software, QÜIKO™, which allows automatic analysis. A mathematic algorithm was developed in order to calculate flight time using total acceleration signal (vector sum of three axis of the accelerometer and information recorded by gyroscopes). Jump height can be calculated with the Bosco, Luhtanen (*Bosco, Luhtanen & Komi, 1983*) equation.

*Force Plate.* The force plate "Twin Plates" (Globus Sport and Health Technologies LLC, Codogné, Italy) ($240 \times 400$ mm) records the data at a frequency of 1,000 Hz with an error of less than 1%. It was used at the same time as the inertial device WIMU™. The

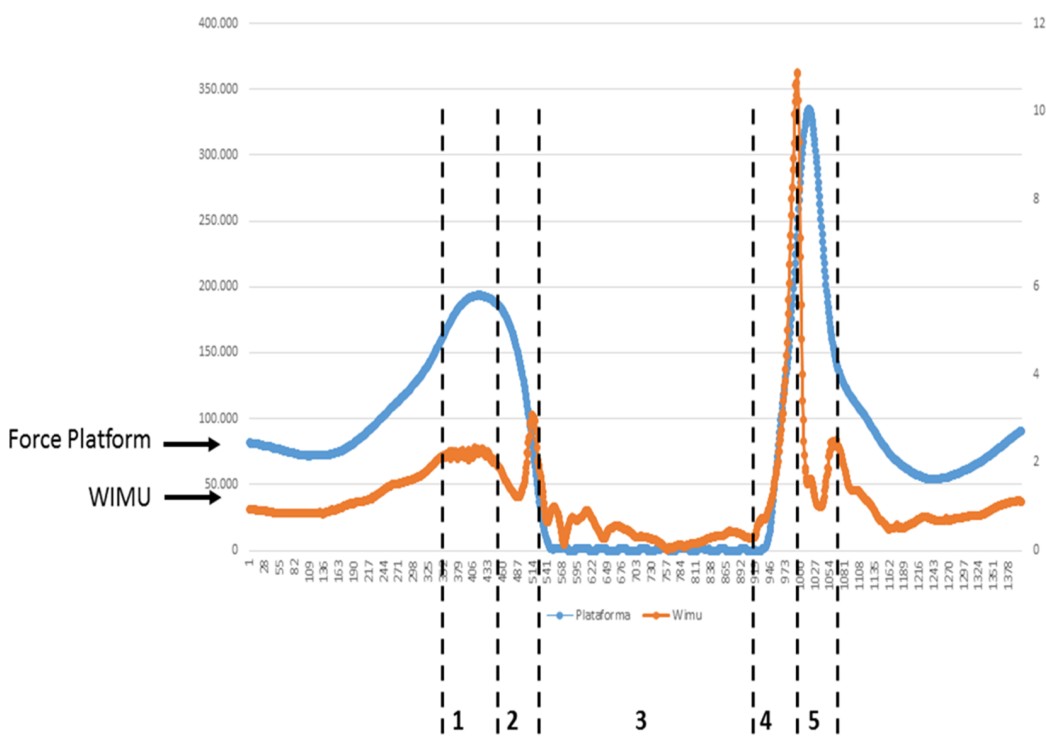

**Figure 1 A graphic example of flight time data of WIMU and force plate.** Squat jump recorded by WIMU and force platform. (1) Knees flexion (concentric); (2) Takeoff; (3) Flight time; (4) Touch-down; (5) Knees flexion (eccentric). The orange line represents WIMU inertial device measured in gs (right axis). The blue line represents force platform measured in Ns (left axis).

plate was connected to a laptop computer for the real-time feedback via the software "Ergo system" (Globus Sport and Health Technologies LLC, Codogné, Italy). Flight time is automatic calculated by the force platform software.

## Statistical analysis

A first descriptive analysis with averages and standard deviation was performed to characterize the sample. The intraclass correlation coefficient (ICC) (2.1) was used for the analysis of reliability of the flight time of the CMJ and the SJ. The coefficient of variation (CV) was used to analyze the reliability of the instrument. The absolute reliability was determined by calculating the indexes *Standard Error of Measurement* (SEM) (SEM = SD√(1-ICC) where SD is the standard deviation of day one and day two) and the *Smallest Real Difference* (SRD) (SRD = 1.96 × √2 × SEM) (*Weir, 2005*). Both SEM and the SRD were calculated in absolute terms and in percentage for easier interpretation and comparison with other measuring devices. Similarly, both the SEM and the SRD became percentages to facilitate comparison with other studies. The percentages were calculated according to the following equation: SEM% = (SEM/mean flight time of the two repetitions) · 100; and SRD% = (SRD/mean flight time of the two repetitions) · 100. All the reliability tests were performed for the force plate and the inertial device. Similarly, the SEM and the SRD are indicators that express the absolute reliability of the device, shown at the same

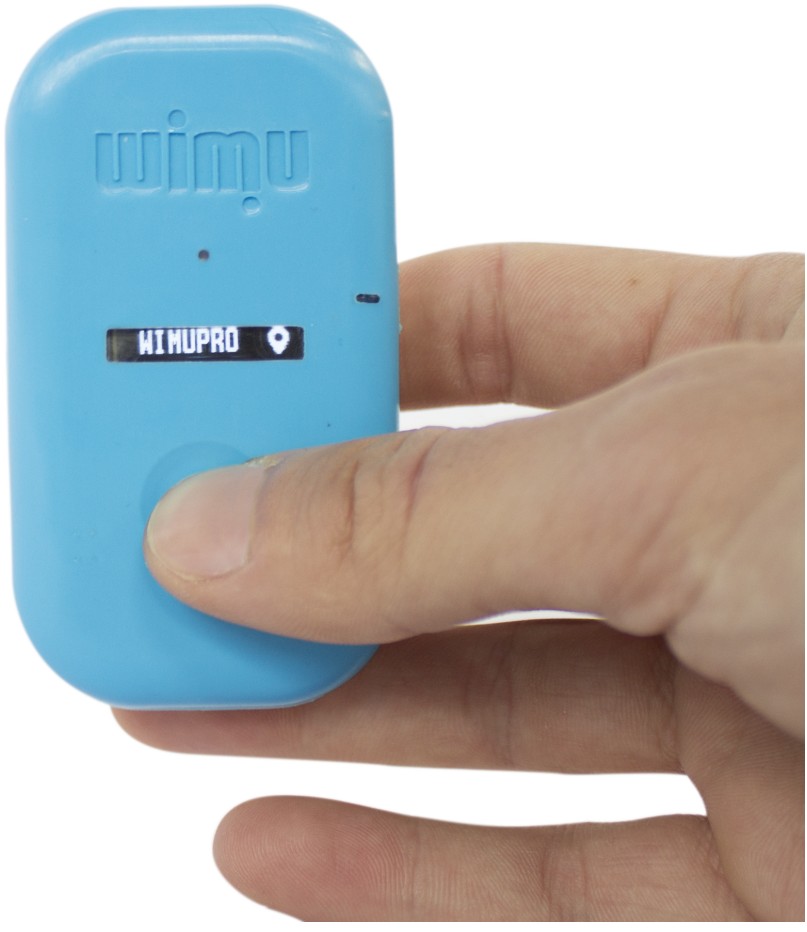

**Figure 2 WIMU inertial device.** Photo credit: Sergio Ibáñez and Javier García.

measurement unit of the instrument. Besides, the SEM results are highly independent from the population under study, not as the ICC (*Weir, 2005*). To extrapolate the results of the study, it has been chosen to express these indexes in absolute terms, thus being able to compare the results with those of other instruments more easily (*Atkinson & Nevill, 1998*).

The calculation of the concurrent validity was performed by using the Pearson correlation coefficient ($r$). This analysis was complemented with the realization of the Bland–Altman plots. This representation indicates the degree of agreement between both instruments, not only the degree of relation (*Bland & Altman, 1986*).

The level of significance was established at $P < 0.05$. All of the analyses were performed by using the statistical package SPSS 21.0 for Windows (IBM Co., Chicago, USA), except for the Bland–Altman plots, which were performed by using the software Graphpad Prism (Graphpad, Inc. La Jolla, CA, USA).

## RESULTS

The descriptive analysis showed that the flight time registered by the inertial device WIMU™ was almost identical than the one registered by the plate (Table 1), but there

**Table 1 Descriptive statistics (Flight time in milliseconds and jumping performance in centimeters), intraclass correlation coefficients, reliability of the inertial device and flight time correlation of the CMJ and the SJ.**

| | | WIMU | GLOBUS | ICC | IC 95% | $r$ | SEM | SEM% | SRD | SRD% | CV% |
|---|---|---|---|---|---|---|---|---|---|---|---|
| CMJ | Time | 436.31 ± 13.70 | 437.62 ± 14.91 | 0.97 | 0.96–0.98 | 0.95 | 9.69 | 2.2 | 26.85 | 6.2 | 3.1 |
| | Performance | 23.34 ± 0.02 | 23.48 ± 0.02 | | | | | | | | |
| SJ | Time | 416.11 ± 10.70 | 416.72 ± 12.50 | 0.96 | 0.94–0.97 | 0.93 | 5.60 | 1.4 | 16.35 | 3.9 | 2.5 |
| | Performance | 21.23 ± 0.01 | 21.29 ± 0.02 | | | | | | | | |

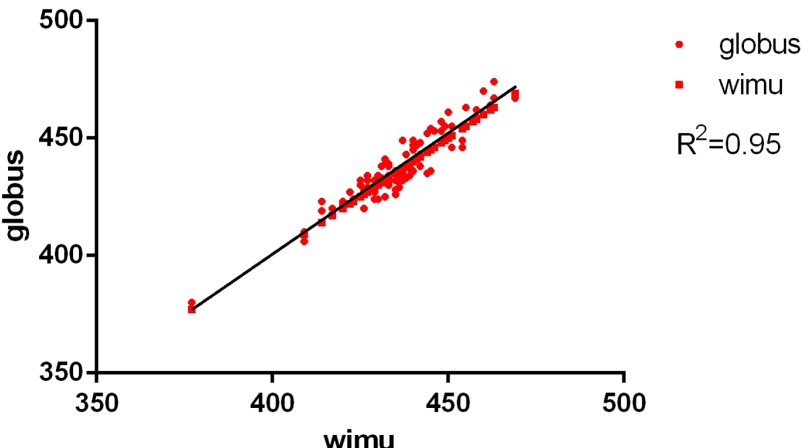

**Figure 3 Concurrent validity between force plate and WIMU: CMJ.** Correlation of CMJ (ms).

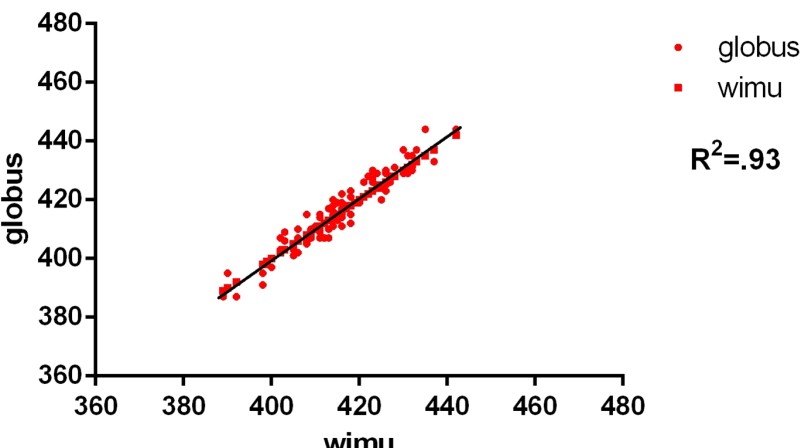

**Figure 4 Concurrent validity between force plate and WIMU: SJ.** Correlation of SJ (ms).

were no significant differences between both devices. The average flight time in the CMJ was: force plate = 437.62 ms; WIMU™ device = 437.31 ms. For the SJ, flight time was: force plate = 416.72 ms; WIMU™ device = 416.11 ms. The results showed an almost perfect relation between the inertial device WIMU™ and the contact platform, both in the CMJ (ICC (2.1) = 0.97, 95% CI [0.96–0.98], $P < 0.001$) as in the SJ (ICC (2.1) = 0.96,

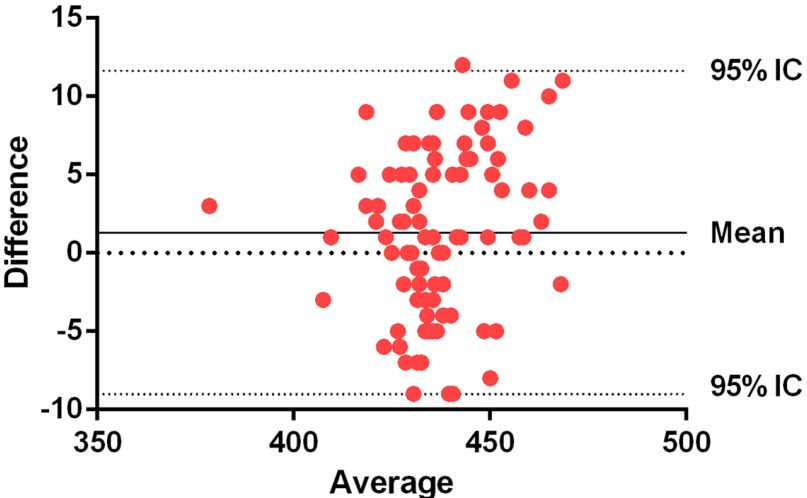

**Figure 5 Bland–Altman plots for force plate and WIMU in SJ.** The central line represents the absolute average difference between instruments. Short-dashed lines represent the upper and lower 95% limits of agreement. Counter movement jump (flight time) (ms).

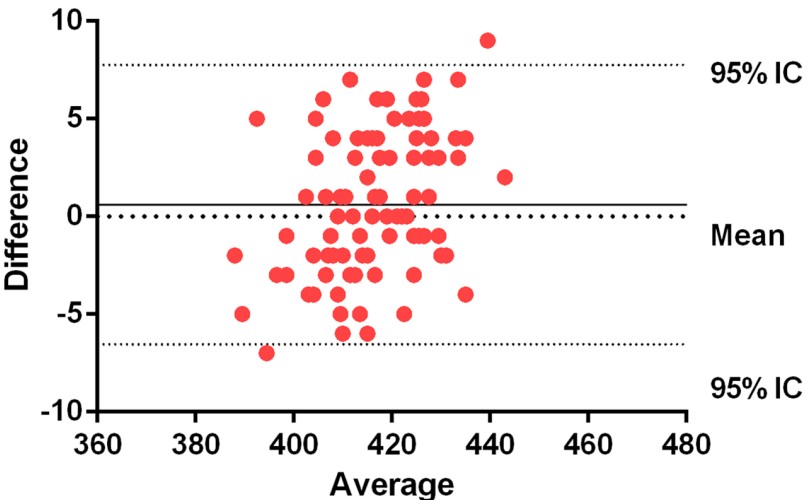

**Figure 6 Bland–Altman plots for force plate and WIMU in CMJ.** The central line represents the absolute average difference between instruments. Short-dashed lines represent the upper and lower 95% limits of agreement. Squat jump (flight time) (ms).

95% CI [0.94–0.97], $P < 0.001$). The reliability analysis showed a SEM% of 2.2% for the CMJ and of 1.4% for the SJ. The SRD% is 6.2% for the CMJ and 3.9% for the SJ. The CVs were very low for both periods of flight time of jump (CMJ: 3.1%; SJ: 2.5%) (Table 1). In addition, the Pearson correlation coefficient is almost perfect in both cases ($r > 0.9$) (Table 1; Figs. 3 and 4).

Bland–Altman plots show an average systematic trend of 1.31 (0.29%) milliseconds between the force plate and the inertial device in the CMJ, and an average systematic trend of 0.61 (0.13%) in the SJ. The trend is the average difference between the two measures. As these values are positive, the force plate gets higher values than the inertial device (Figs. 5 and 6).

## DISCUSSION

The objective of the present study was to evaluate the validity and reliability of the WIMU™ device for the assessment of the flight time of SJ and CMJ. The results show the high concurrent validity and reliability that the WIMU™ device presents compared with a force plate. The WIMU™ device has proven to be a useful instrument. The results from this study conclude that WIMU™ provides similar flight times in SJ and CMJ to the criterion method.

In previous studies, correlation coefficient has been the main method to assess validity and reliability. It has been suggested that other methods, as Bland–Altman plots, provide more relevant information about agreement between two measures (*Bland & Altman, 1986*). For example, it has been probed that Myotest and Optojump have a high correlation according to ICCs (0.98), but with a systematic bias of around 7 cm in vertical jump (*Casartelli, Müller & Maffiuletti, 2010*). In fact, a constant error will not be detected on the correlation analysis and, therefore, it cannot be concluded that devices are accurate (*Magnúsdóttir, Orgilsson & Karlsson, 2014*). The data presented in the Bland–Altman plots show that most of the jumps are close to the media of the differences between instruments in both jump modalities, showing a high level of agreement (*Bland & Altman, 1986*) and a high correlation between them. The ICC presents a great precision of measurements (>0.93). Similarly, by analyzing the reliability of the devices, excellent CVs (<10%) can be appreciated (*Choukou, Laffaye & Taiar, 2014*; *Atkinson & Nevill, 1998*). The SEM% shows very low percentages of absolute error (*Collado-Mateo et al., 2015*).

Several studies have examined the validity of different methods of analysis of the vertical jump in comparison to force plates. Previous studies have misreported the vertical jump performance, found discrepancies between devices compared with force plates of around 10 cm measuring flight time (*Aragón, 2000*; *Moir, 2008*) or measuring jump height, from recorded flight time, in contact mats (*Aragón, 2000*; *Buckthorpe, Morris & Folland, 2012*). It has also been reported that mastery of the jumping technique may affect the jump performance (*Buckthorpe, Morris & Folland, 2012*). On the other hand, contact mats measure flight time according to the moment when the subject leaves the ground, and miss some data of the initial rise of the center of mass before the take-off (*Buckthorpe, Morris & Folland, 2012*). Also, the differences could be due to the different ascending and descending phases and landing (*Aragón, 2000*). In fact, some studies report as limitation the lack of gyroscope that can detect the inclination of the body in take-off and landing (*Casartelli, Müller & Maffiuletti, 2010*; *Requena et al., 2012*).

Studies have shown lower values for the validity (ICCs scores), of their devices to those found with the WIMU™. Choukou, Laffaye (*Choukou, Laffaye & Taiar, 2014*) studies the validity of the Myotest in comparison to a force plate, finding ICC values between 0.86 and 0.96. Casartelli, Müller (*Casartelli, Müller & Maffiuletti, 2010*), by studying the same device, finds similar results. The analysis of reliability shows very low CVs, better than those found when the reliability of mobile applications was developed (*Gallardo-Fuentes et al., 2015*), photoelectric cells (*Glatthorn et al., 2011*), or other inertial devices such as accelerometers (*Casartelli, Müller & Maffiuletti, 2010*;

*Choukou, Laffaye & Taiar, 2014*). When the results are compared to the ones obtained with high-speed cameras, the results are practically the same (*Requena et al., 2012*). Force plates are considered the most accurate tools for measuring the vertical jump, since they allow identifying the moment of take-off with great precision (*Requena et al., 2012*; *Glatthorn et al., 2011*; *Enoksen, Tønnessen & Shalfawi, 2009*). This device, force platform, is enable to measure, in eccentric and concentric phase of the movement, the force and power production (*Buckthorpe, Morris & Folland, 2012*), allowing more detailed analysis of subjects' training. WIMU$^{TM}$ allows providing a similar analysis of force plates (as results concluded) and, contrary to force plates, WIMU$^{TM}$ still records data of more variables during flight time (such as G force in take-off and landing or inclination, among others), which makes the analysis more detailed. These results demonstrate the ability of the WIMU$^{TM}$ system to play the measurements that a force plate makes, having the advantage of being lightweight and portable. These measurements are valid for both the CMJ and the SJ. Due to its properties of size and weight, this device can be easily placed in any segment of the body to measure the vertical jump (center of mass, hips, back, lower body, etc.). The data collection is done with a single computer wirelessly connected to the device, so a great time preparing for the test is not needed. In addition, subjects do not have to be connected by any cable or have to take-off and land in a delimited area. Moreover, thanks to the specific software, the device provides immediate feedback.

## CONCLUSION

Because of the ability of this device to collect data on different types of jump and the relevant information from these, WIMU$^{TM}$ is a valuable tool for controlling the training and competition. Throughout control of the jumps, changes produced by a particular training program can be detected, being able to redirect or modify it according to the results (*Casartelli, Müller & Maffiuletti, 2010*). It can also be used to determine the fatigue accumulated during the same work, adapting the breaks between sets for a full recovery (*Sato, Smith & Sands, 2009*; *Haff et al., 2003*). The WIMU$^{TM}$ device does not need cables, so it greatly facilitates its placement in the subject's body as well as the freedom of movement.

### Funding

This work has been partially supported by the "Ayuda a los Grupos de Investigación (GR15122)" of Govern of Extremadura (Economy and Infrastructures Department), with the support of the European Union through FEDER funds. There was no additional external funding received for this study. The funders had no role in study design, data collection and analysis, decision to publish, or preparation of the manuscript.

### Grant Disclosures

The following grant information was disclosed by the authors:
Ayuda a los Grupos de Investigación: GR15122.

## Competing Interests

José Pino, PhD is a Sport Science advisor at RealTrack™. To ensure the independence of the analysis and the article, this author made no contribution to the data analysis and results section. Instead, he contributed significantly in the other parts of the manuscript without having access to the data set or data analysis.

## Author Contributions

- José Pino-Ortega conceived and designed the experiments, performed the experiments, authored or reviewed drafts of the paper, approved the final draft.
- Javier García-Rubio performed the experiments, analyzed the data, contributed reagents/materials/analysis tools, prepared figures and/or tables, authored or reviewed drafts of the paper, approved the final draft.
- Sergio J. Ibáñez performed the experiments, analyzed the data, contributed reagents/materials/analysis tools, authored or reviewed drafts of the paper, approved the final draft.

## Human Ethics

The following information was supplied relating to ethical approvals (i.e., approving body and any reference numbers):

The ethics committee of the Universidad de Extremadura granted ethical approval to carry out the study (nº Registro 67/2017).

## Data Availability

The raw data are provided as Supplemental Files.

## Supplemental Information

Supplemental information for this article can be found online at http://dx.doi.org/10.7717/peerj.4709#supplemental-information.

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
