# Peer review of "Validity and reliability of the WIMU inertial device for the assessment of the vertical jump"

_PeerJ, doi:10.7717/peerj.4709_

## Round 0.1 · original submission · Major Revisions

The three reviewers and I see the potential in this manuscript, but also a number of limitations that need to be addressed. In particular, reviewer two and I have some reservations regarding aspects of the methodology which read a little bit like an advertorial. While the potential conflict of interest was included in the submission, it does not appear that the reviewers can access this information. Regardless of this, it is imperative that the manuscript is written as objectively as possible and you refrain from making it sound too much like you are involved in the company, even though you are. Perhaps look at some of the papers published on apps such as the MyJump or MySprint by the app developers or even by Brett Contreras and his studies on the hip thrust to get an idea of how people intimately involved in a product or an exercise can still write more objectively in their manuscripts.

Reviewer 1 ·

Basic reporting

The study is well designed and the manuscript is generally well structured. The results indicated the validity and reliability of the inertial device WIMU for two different type of vertical jumps. The paper focuses on a topic which should be of interest to coaches, trainers, physical educators, and sports scientists, and has relevance to exercise training. However, I have some reservation that the practical impact of the work may be limited and that the work has limited novelty despite the limited research in this area. Overall, I recognize the strong scientific merit of the research, which uses a good research design and robust methods of measurement and analysis.

Experimental design

The purpose of the study is investigate validity and reliability of the inertial device WIMU for two different type of vertical jumps. The authors used test and retest method to evaluate reliability of the data. At the same time, authors also used the force platform as a reference to investigate the validity of WIMU.

Validity of the findings

The statistical analysis was also support the purpose of the study. Intra class correlation, coefficient of variation, standard error measurement were used to evaluate the reliability and validity of WIMU device. The statistical analysis support the purpose of the study. However, there are few improvement need to be made that listed below in the comments section.

Additional comments

The study is well designed. There are a few areas need to be improved. The specific comments below highlight areas where improvement is recommended although these are relatively few because of the good quality of design and analysis. Overall, this paper presents work of good quality but there remains a significant question on how much new knowledge the work brings to the field. Evaluation of potential impact is a difficult challenge and it can be difficult to predict the impact (in terms citations) of a paper before it reaches the public domain. Ultimately this should be a decision for the editor.

Abstract: Please provide the full name of CMJ and SJ as they introduced at the first time.
ln 136: how did you define flight time from force platform? What is threshold? Since using the force platform why not calculate jump height from integration of impulse? Any filter being used for signal processing from the two instruments?
ln 146: please check typo CCI.
ln 151: please explain what is mean power spectrum of the two repetitions? Power spectrum analysis?
References: please provide completed page number of articles (reference 6 and 27)

Reviewer 2 ·

Basic reporting

Overall, this is a well-designed study that aims to evaluate the validity and reliability of the WIMU device. Despite the effective design of the study, the manuscript has a fatal flaw, in that all of the sections about the WIMU or QUIKO reads as though it is an advertisement for RealTrack systems. There are numerous sections that don't really serve to give information relevant to the study, rather it serves to inform the reader of the neat features and the advantages that WIMU and QUIKO has over other technologies.

In fact, after finishing the methods section, I had to go back into the text to make sure that none of the authors had a conflict of interest with RealTrack. I did some Googling, and it turns out the lead author lists himself as being employed by RealTrack systems on Linkedin. While I cannot see the acknowledgments section in my version of the manuscript, if the first authors' LinkedIn profile lists this as his employment, it really should be included as an affiliation in a manuscript. This conflict of interest (that should have been disclosed as an affiliation), has obviously bled into the authors' writing, which is a significant flaw with this manuscript.

I have chosen not to address the rest of the paper until the "advertorial" writing issues have been cleared up.

Experimental design

Did not assess.

Validity of the findings

Did not assess

Additional comments

Here are the major issues that I came across in the writing. I chose not to read any farther than the methods section.

Line 64-71 This section could have been accomplished in a single sentence. Instead, the authors chose to list off neat features of the device, and how much better it is than other types of technologies. This particular section reads like an advertisement.

L117-130 Only about half of this information is actually useful to the reader for the purpose of informing or allowing for replication. The other half of the information reads like a bullet-pointed list in a product pamphlet of all of the customization options for the Quiko software. We do not need to know, as a reader, about the adjustment parameters ("features"), unless you are providing this information so that we can replicate or better understand your methods.

·

Basic reporting

The research havea real interest and deser to be published. However, the author should disus about parameters and device potential that has not been investigated inn the present study. Jumping performance needs to be introduced in the results and needs to be discussed.

Experimental design

The research is original and within scope of the journal. The research question is fairly well defined and relevant. However, the analysis of the results only focus on flight time, while jumping performance (in meters) is never considered. Methods used to analyse the validity and the reliability seems to be appropriated and complete.

Validity of the findings

Findings should be implemented with data on jumping performance. Moreover, in many cases the authors discus about parameters of the device that have not been investigated. These parameter are out of subject and caution is really needed as they haven’t been investigated.

Moreover, the authors have to be aware that they can compare their flight time results with only the studies that investigated that particular parameter.

Conclusion is too long and should focus on the research hypothesis.

Additional comments

Abstract :
Why don’t you gather all reliability analysis at the same time ? You introduce ICC L25 then you introduce CV and SEM L28-29.
L34. Remove “on any surface” as it hasn’t been investigated.

Introduction :
L40-L43. Out of subject and not necessary.
L60 The authors should explain in more detail why and how jumping performance can be estimated through the measurements of flight time. References are also needed here.
L65-L70. This is the place where authors could explain in more details all potential possibilities of the WIMU system. After the general presentation the authors should exclusively focus on the jumping measurement.
L68-69. Why the fact that WIMU system receive data during the jump could be an advantage ass we only take into account the flight time ? Moreover, with accelerometer, it is much more difficult to detect accurately the moment where the subject leave the floor and when he lands. The argumentation of the authors need to be changed. If the data measured during the flight phase with the WIMU system are used to improve the estimation of flight time and jumping performance, please explain how !

Material and methods
L89. Wimu installation on the back need more details.
L105. Was it the same examiner ?
L113 : what was the instruction for landing ?
L122 : does the WIMU contain any Gyroscope ?
L140 and L144. Why do the authors separate the two reliability analysis ?
L151 and 152. What does mean “_” in the formulas ? Not clear.

Results
Jumping performance results also need to be presented.
L177 and L178 : could you check your data ? Why in CMJ a superior average systematic tends leads to a lower % ?

Discussion
L197-198. Give an example. (comparison betwwen Myotest and Optojump ?).
L222 : ”this device is enable to… ” which device ? Th WIMU ? It is not clear.
The authors should discuss the importance of the landing with all device measuring the jumping performance through flight time !
The discussion should include the results on the jumping performance. Moreover, the authors have to be aware that they can compare their flight time results with only the studies that investigated that particular parameter.
L226. The authors discuss about the ability of the WIMU system to measure other parameters like force. They need to be more prudent and to use conditional as they didn’t investigate this parameter. Moreover, a short overview on the figure 1 seems to show that values are not the same from WIMU to the force plate.

Conclusion
Conclusion is too long and should focus on the research hypothesis.
L250. The authors could present other potential advantages of the system but have to mention that investigations are needed to verify their reliability.

References
Please check all references format

Tables
As ICC is a reliability parameter. Why don’t you introduce it in the table 2 ?
Table should also include Jumping performance (in meters)

Figures
Figure 1 : what is the measured parameter ? Present the unit.
Figures 3 and 4 : Please introduce the linear regression and the R². Please introduce the unit.
Figures 5 ans 6 : please introduce the unit.

---

## Round 0.2 · Major Revisions

We thank the authors for taking on board many of the initial comments from the three reviewers. Please further pay particular attention to the comments from several reviewers regarding what measures have been shown to be valid and reliable (flight time) and ensure your conclusions match what your data actually demonstrated. Also, please pay particular attention to reviewer 2's comments about the standard of academic writing. As reviewer to have suggested, it might be worthwhile to access the assistance of someone who can assist you in this matter.

Reviewer 1 ·

Basic reporting

The authors revised the previous errors. The results indicated the validity and reliability of the inertial device WIMU for counter movement and squat vertical jumps. Overall, I recognize this research, which uses a good research design and robust methods of measurement and validation analysis.

Experimental design

The purpose of the study is investigate validity and reliability of the inertial device WIMU for two different type of vertical jumps. The authors revised the errors and typos. Test and retest method to evaluate reliability is appropriate for the study.

Validity of the findings

The statistical analysis was also support the purpose of the study. However, in the conclusion part. the authors indicated that the new device is valuable tool for controlling the training and competition. This may not be answered by this study. There are numerous factors that would influence training and competition outcomes other than vertical jumps. Long term study may be needed. Therefore, the authors should consider to rewrite this senescence.

Reviewer 2 ·

Basic reporting

Overall, this has been a big improvement over previous. There are still numerous areas to be improved, and another advertorial section to tone down in the discussion. The authors need to be careful of over-stretching the conclusions they draw from this study. Flight time measured by the WIMU is valid and reliable, but we have no data to suggest anything about other measures that might come from the WIMU, thus we cannot conclude anything about the device quality overall, just about its measurement of flight time.

Finally, having a native English speaker would be helpful for dealing with some if the awkward and confusing language throughout. I have tried to point out the worst of them, but I certainly didn’t get them all.

Experimental design

Design is organized well.

Validity of the findings

The findings are logical and fine. The conclusions are not- the authors found that flight time, and only flight time, is valid and reliable, yet write about the device as though it has been validated for many other variables. They seem to be really stretching the scope of their findings.

Additional comments

Abstract
Which CV did you use? Hopkins’ spreadsheet?
Do you mean that you used standard error of the measurement,
T, rather than standard error of the means? These are not the same thing.

L39 please check this for grammar

L46 please change ‘measured’ to “estimated”.
L49 “jump height is an indirect method to jump height”...? Please check this sentence for accuracy.
L50 different methods for calculating what exactly?
This error of 2.2 cm- under what conditions and when using which devices does this error occur?
L75 what do you mean by early stages?
L77 what kind of training? Sport practice?
L90 with HANDS on their hips
L98 how much rest did each participant get between trials?
L116 can you explain more about the specific statistic you’re referring to when you say you calculated CV
L143-146 ICC only tells you about the reliability of the device- not the relationship between devices- the Pearson correlation does. Please modify accordingly.
L159 are you sure that you mean misreported? This rd doesn’t make sense in the context of the rest of the sentence.
L186-188- I don’t think you are referring to contact mats still- please add a phrase in this sentence that your focus is on the accelerometer.
L199-210 And here we are again with the advertorial- in this study you ONLY tested flight time, this you cannot talk about how great the other features are. You specifically suggest that the WIMU can make the same measurements as a force plate, then state in the next sentence that these measurements are valid. You don’t have multiple measurements/variables that are valid- you have ONE. A sentence or two about how this type of technology has advantages over force plates is fine, but you have here another entire paragraph about hoe great the WIMU is...
L217 Neither of these two studies support your statement- these two studies measured barbell kinetics/kinematics, not flight time as was measured in this study.

·

Basic reporting

L43 : a sentence explaining that accelerometer coul be sucessfull to measure jumping performance is missing here (maybe with a reference to another system)
L47 : replace "the more" by "an"
L49 : I suppose you mean "Also, the Fight time is an indirect technique to measure jump height or muscle power ...
L50 : "suggested"(?)
L54 : You need here to describe that one function of the WIMU is to assess jumping performace throught the measurement of flight time.
L58 : replace inthe by in the
L90 : "Once in position, the subjects were instructed (approximately a 90° angle) as quickly as possible and then jump as high as possible." What are they instructed to do AQAP ? Not clear.
Table 1 : Insert standart deviation for jumping performance + units
L199-206 : As you did'nt compared force signals of both systems in the present study, you cannot affirm that WIMU allows providing a similar analysis of force plate. In this paragraph, you need to be more prudent in your assertions. Conditional should be used in the otentioal of the WIMU. However you can write that it need to be verufy by further investigations.
L204 "These measurements are valid for both the CMJ and the DJ." => The only measurement that has been demonstrated to be valid in your study is "flight time"
Conclusion : You research is on flight time validity and reliability and not on training control. I invite the authors to keep the conclusion around the research hopothesis. They can of course suggest the need of additional researches in order to investigate te complete potential of WIMU system.

Experimental design

No comment

Validity of the findings

No comment

Additional comments

The manuscript has been improved but a few change are still needed before acceptance. All the text need to be carefully ready to remove all typo errors.The potential of the WIMU could be in the discussion, but conditional tense should be used and authors have to be aware that they can not assert things that has not already been investigted. But they can suggest a need for further studies. The conclusion still needs improvements

---

## Round 0.3 · Minor Revisions

Reviewer one has one small comment for you to respond to. Please respond to this comment so that we can accept this manuscript for publication.

Reviewer 1 ·

Basic reporting

Authors made corrections based on the previous review. However, there is one part still need to improve.
From Ln 182 to Ln 186. The variable from study was flight time based and only one variable was investigated. The rest of the variables were not being analyzed (compare to other reference), so you can only indicate the reliability of flight time compare to force plate. The reliability of the other variables were not being evaluated in this study. So, you may consider to change the statement.

Experimental design

The design of the study was good

Validity of the findings

Validity of the study was confirmed based on the comparison of flight times from
two different instruments.

Additional comments

From Ln 182 to Ln 186. The variable from study was flight time based and only one variable was investigated. The rest of the variables were not being analyzed (compare to other reference), so you can only indicate the reliability of flight time compare to force plate. The reliability of the other variables were not being evaluated in this study. So, you may consider to change the statement.

Reviewer 2 ·

Basic reporting

The authors have made a substantial effort toward improving this manuscript. They should be commended for their immense improvements. Only a few comments related to writing.

Experimental design

Still good.

Validity of the findings

The authors have limited the scope of their conclusions enough so that they appropriately reflect the findings of the study.

Additional comments

L175 Please change "really" to "very"
L233 Please change "thought" to "throughout"

·

Basic reporting

The authors answered successfully to the reviewer's comments. From my point of view, the article is now acceptable.

Experimental design

The authors answered successfully to the reviewer's comments. From my point of view, the article is now acceptable.

Validity of the findings

The authors answered successfully to the reviewer's comments. From my point of view, the article is now acceptable.

Additional comments

The authors answered successfully to the reviewer's comments. From my point of view, the article is now acceptable.

---

## Round 0.4 · Minor Revisions

We thank the authors for their willingness to improve the paper but I believe there are some small areas within the last revision that need final clarification.

Line 182 – 183: unless I am mistaken, contact mats also measure flight time. Therefore this sentence is unclear with respect to the agreement between inertial sensors, force plates and contact mats in measuring flight time. Please make this clearer.

Line 183: this sentence should read "It has also been reported ...."

---

## Round 0.5 · accepted · Accept

We thank the authors for making the required amendments to the manuscript which allows us to accept it for publication